# Trends and Factors Associated with HIV Testing among Adolescent Girls and Young Women in Lesotho: Results from 2004 to 2014 Lesotho Demographic and Health Surveys

**Onalethata Ntshadi Sonny and Alfred Musekiwa ***

School of Health Systems and Public Health, Faculty of Health Sciences, University of Pretoria, Pretoria 0002, South Africa
* Correspondence: alfred.musekiwa@up.ac.za

**Abstract:** HIV/AIDS is prevalent among adolescent girls and young women (AGYW) in Lesotho, and among the top five infectious diseases causing a high mortality rate in Africa. The paramount management of HIV is based on screening, prevention, and therapeutic interventions, of which HIV testing and counselling (HTC) is the gateway. The utilization of HTC services among AGYW is limited owing to numerous barriers encountered by this population group. The aim of this study was to assess trends and factors associated with HTC among AGYW in Lesotho. A secondary data analysis was used to analyze data sets extracted from the 2004, 2009, and 2014 Lesotho Demographic Health Surveys (LDHS). The LDHS was conducted using a cross-sectional study design and samples drawn were representative of the whole population of Lesotho. We used descriptive analysis to determine trends in HTC. In determining factors associated with the uptake of HTC, univariate and multivariable logistic regression models were applied on the 2014 LDHS. All analyses were adjusted for unequal sampling probabilities using survey weights. The number of AGYWs analyzed were 2743 in 2004, 2393 in 2009, and 2842 in 2014. The overall prevalence estimates of HTC uptake were 12%, 62.2%, and 72.5%, respectively. For the 15–19 years AGYW, HTC uptake rose from 6.2% (2004), 46.3% (2009), to 57.9% (2014), while for the 20–24 years age group, the rates were 18.7%, 80.2%, and 88.3%, respectively. For the 2842 AGYW in 2014, the odds of ever having an HIV test were significantly higher for those aged 20–24 years (aOR 2.15, 95% CI 1.61 to 2.87, $p < 0.001$), in a union (aOR 3.21, 95%CI 2.25 to 4.58, $p < 0.001$), with Mother-to-child transmission of HIV (MTCT) knowledge (aOR 1.53, 95%CI: 1.21 to 1.94, $p < 0.001$), with HIV non-discriminatory attitudes (aOR 2.50, 95%CI 1.87 to 3.34, $p < 0.001$), and those who had ever been pregnant (aOR 11.53, 95%CI 7.46 to 17.84, $p < 0.001$). HTC uptake among AGYW in Lesotho is below expected targets, hence we recommend optimizing access to HTC services, especially for AGYW aged 15–19 years.

**Keywords:** HIV testing; AGYW; DHS; trends; factors; Sub-Saharan Africa; Lesotho





## 1. Introduction

Adolescent girls and young women (AGYW) in Sub-Saharan Africa are disproportionately affected by the HIV/AIDS disease. Although they comprise only 10% of the total population in the region, females aged 15–24 years in Sub-Saharan Africa accounted for 25% of all new infections in 2020 [1]. The HIV incidence rate of their male counterparts, who equally constitute 10% of the total population in Sub-Saharan Africa, was estimated at only 8% [1]. Globally, the probability of AGYW aged 15–24 years to be living with HIV is double compared to their male counterparts [2]. In 2021, UNAIDS estimated that there were approximately 38.4 million people living with HIV (PLHIV) worldwide, with two-thirds (25.6 million) from the African region [2,3]. Of these two-thirds, 63% were women and girls [2]. Amongst the countries hugely affected by the HIV epidemic is Lesotho, with 21.1% of persons aged 15–49 years living with the disease [4]. Of these PLHIV in Lesotho, 61% are

women. HIV prevalence disparity by gender is particularly evident among young people worldwide, and in 2020, 9% of AGYW were living with HIV in Lesotho compared to 3.9% of their male counterparts [4]. There were approximately 6700 adults (ages 15–49 years) newly infected with HIV, of which 41.8% were young people aged 15–24 years [4]. Although HIV testing is an effective prevention strategy, many individuals are still not utilizing HTC services, with a global estimation of about 5.9 million people living with HIV but unaware of their HIV status [3]. UNAIDS recognizes sex workers, including AGYW, as key population for HIV/AIDS. According to UNAIDS [3], female sex workers have a 30 times higher risk of acquiring HIV than adult women in the general population. It is evident that AGYW are immensely affected by HIV infection more than males in the same age group. Numerous factors contribute to this disparity, including gender inequality, cultural norms, socio-economic position, geographical location, age, educational status, sexual orientation, and the biological structure of women's reproductive system [5–7]; women have a large bare mucosal surface of the vagina, and the cervical cells exhibit an elevated presence of HIV co-receptors compared to the foreskin cells in men, which therefore leads to ease of the acquisition of HIV infection by women [5].

It is essential to implement HIV testing on a wide scale among AGYW to achieve a significant public health impact; however, this has proven challenging to accomplish [8]. Saul et al. [8] report that though there has been a significant improvement in HIV testing among AGYW following the Determined, Resilient, Empowered, AIDS-free, Mentored and Safe (DREAMS) initiation, challenges remain for higher coverage. DREAMS was initiated in 2014 by the United States President's Emergency Plan for AIDS Relief (PEPFAR) with the aim to help countries to achieve epidemic control of HIV/AIDS by targeting AGYW [8]. Many countries failed to achieve Joint United Nations Programme on HIV/AIDS (UN-AIDS) 90–90–90 targets set for 2020, with the Lesotho AGYW population group scoring 84–51–89 in 2018 [9]. These ambitious targets have now been adjusted to 95–95–95 and set to be achieved by 2030, proposing that 95% of PLHIV will know their status, 95% of all people knowing their HIV status will be on antiretroviral therapy (ART), and 95% of PLHIV on ART will be virally suppressed by 2030 [10]. The UNAIDS HIV prevention cascade score is currently ranked at 85%–85%–68% globally [3]. Almost all countries offer HTC services freely, but the uptake of HIV testing is influenced by individual viewpoint, structural, and societal factors [11]. This study analyzed LDHS data to establish trends and identify factors associated with uptake of HIV testing among AGYW in Lesotho. The findings of the study could be helpful in improving existing AGYW HTC programs.

## 2. Materials and Methods

This study is a secondary data analysis of the three consecutive Lesotho DHS data sets from 2004, 2009 and 2014. The LDHS is a nationally representative, population-based, cross-sectional survey. The survey was conducted across all 10 districts of the country and samples were drawn from both rural and urban areas.

### 2.1. Study Population and Sampling

All three surveys utilized a sample based on the population census and a stratified, two-stage cluster random sampling. The first stage used enumeration areas as the sampling units, whilst the second stage used households. During the first stage, random selection of enumeration areas both in urban and rural areas were carried out. In the second stage, all households in the selected enumeration areas were listed and then randomly selected. Men (15–59 years) and women (15–49 years) who had stayed in the selected households the night before the survey, including visitors, were eligible to be interviewed. Further details on the survey sampling are available on LDHS manuals for 2004, 2009 and 2014 [12–14]. Household sample size varied for each survey, with 2004, 2009 and 2014 having 9892, 10,941 and 9942, respectively. In this study, eligible participants were females aged between 15–24 years old, who were resident in Lesotho during the time of DHSs. The subjects must

have participated in the LDHS for 2004, 2009, and 2014. The exclusion criteria were eligible AGYW who did not respond to questions in the HIV/AIDS section.

## 2.2. Measurements and Outcome

The main outcome of interest was HIV testing measured on the basis of whether one had ever been tested for HIV. In the survey, participants were asked whether they had ever had an HIV test, and the responses were binary (yes or no). We treated 'I don't know' or no response as missing. The LDHS used qualified interviewers and structured questionnaires to collect data on many variables related to HIV/AIDS. Such variables entail HIV knowledge, attitudes towards PLHIV, risky sexual behavior, and HIV testing. The HIV testing results were linked with the sociodemographic data collected in the surveys. In determining trends in HIV testing among AGYW in Lesotho, we used data sets from the three sequential 2004, 2009 and 2014 LDHS.

To determine potential factors associated with HIV testing among AGYW in Lesotho, we only used the data set from 2014 LDHS. The predictor variables included in the study were socio-demographics: classified age in years (15–19 and 20–24), employment in the last 12 months (yes or no), level of education (no education, primary, secondary, and higher than secondary), marital status (never married/in union, married/in union, formerly married/in union), place of residence (rural or urban), wealth index (Poorest, poorer, middle, richer, or richest). Wealth index is a multi-pronged measure of household living standard. The score is based on number and the kind of selected assets that a household has. In the LDHS, selected assets included bicycles or cars, television, and housing characteristics, such as drinking water, toilet facilities, and flooring materials. A principal component analysis, where individual households were positioned on a continuous scale of relative wealth, was used to generate the wealth index. The risk behavior predictor variables included lifetime number of sexual partners (1/2/3 or more), had an STI (yes or no), and condom use at last sexual intercourse (yes or no). Respondents who never had sexual intercourse were excluded when quantifying risky sexual behavior. Another imperative independent variable with multiple questions was about knowledge of HIV, and consisted of comprehensive HIV knowledge, knowledge of MTCT and HIV non-discriminatory attitudes, of which the responses were all binary (yes or no). Comprehensive HIV knowledge and HIV non-discriminatory attitude variables were generated by collating responses to some questions in the LDHS data. Comprehensive knowledge referred to knowing that HIV infection can be prevented by using condoms during sexual intercourse and having just one uninfected faithful partner, comprehending that a healthy-looking person can have HIV, and rejecting the most common fallacies about HIV transmission, which in Lesotho are that a person can become infected when sharing food with an HIV-infected individual and by mosquitoes. Knowledge of MTCT was assessed by whether the respondent knew that mother-to-child HIV transmission can occur during pregnancy, delivery, and by breastfeeding. The history of pregnancy and gender-based violence variables were also added, and individuals were asked if they had ever been pregnant (yes or no), or had ever experienced gender-based violence (yes or no). There was no direct question on gender-based violence, but some questions referring to the latter were used to generate the variable. The pooled questions were closely related to gender-based violence and are "Beating justified if wife goes out without telling husband", "Beating justified if wife neglects the children", "Beating justified if wife argues with husband", "Beating justified if wife refuses to have sex with husband", "Beating justified if wife burns the food". Response to these questions was binary (yes or no).

## 2.3. Data Collection and Analysis

This is a quantitative secondary data analysis study; we utilized Lesotho DHS data from 2004, 2009, and 2014 to analyze trends in HIV testing. The latest 2014 survey data set was used to determine factors associated with HIV testing.

Descriptive statistics were applied in quantifying trends in HIV testing using the three successive LDHS data sets. The relationship between HIV testing and predictor variables

was computed and conveyed as categorical variables, with results given in percentages. Predictor variables comprised age in years, employment, marital status, level of education, place of residence, wealth index, lifetime number of sexual partners, condom use at last sexual intercourse, had an STI, comprehensive HIV knowledge, knowledge of MTCT, non-discriminatory attitudes, pregnancy history, and gender-based violence. In determining factors associated with HIV testing, we used univariate/bivariate analysis and multivariable logistic regression model. Using manual backward elimination with a *p*-value < 0.1, independent variables were selected into the multivariable model following univariate analysis of each independent variable against HIV testing outcome. *p*-value < 0.05 was used for the final multivariable logistic regression model. Results were presented as crude odds ratios (OR) and adjusted odds ratios (aOR), with corresponding 95% confidence intervals (CI) and *p*-values. A complete case analysis was performed where cases with missing data on any variables in the multivariable model were omitted from the analysis. No imputation was done, and it was assumed that data were missing completely at random. The primary statistical software for data analysis was STATA version 14, and Microsoft Excel was utilized in creating graphics.

## 3. Results

### 3.1. Patterns in the Trend of HIV Testing from 2004 to 2014

Using Lesotho DHS data from 2004, 2009 and 2014, we investigated trends of HIV testing among AGYW. Overall, a remarkable increase in HIV testing uptake was observed from 2004 to 2014—12% in 2004, 62.2% in 2009, to 72.5% in 2014. It was highest among AGYW aged 20 to 24 years—18.7 % in 2004, 80.2% in 2009, to 88.3% in 2014. HIV testing among AGYW aged 15 to 19 years remained low and slightly increased over the period—6.2% in 2004, 46.3% in 2009, to 57.9% in 2014. For both age groups, the pattern indicates an abrupt increase in HIV testing from 2004 to 2009, followed by small increase from 2009 to 2014. (Figure 1).

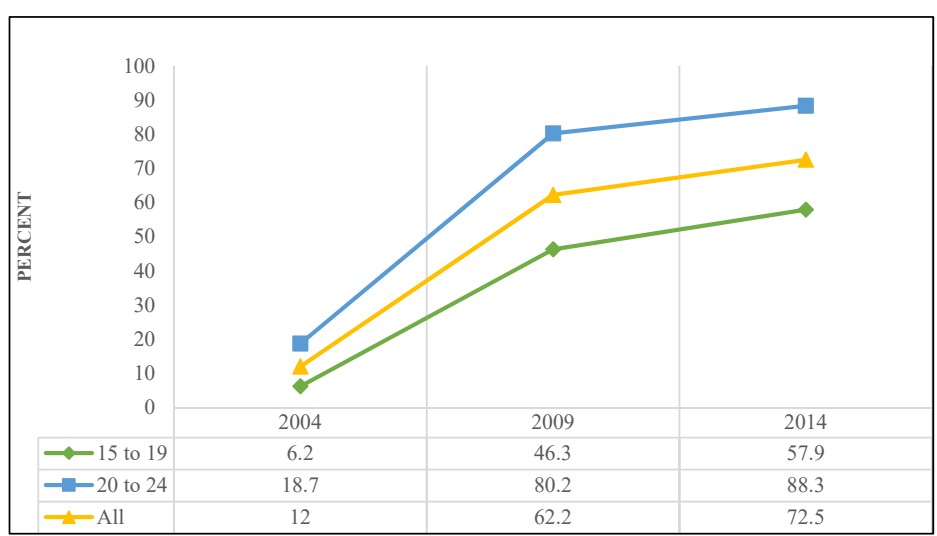

**Figure 1.** Trends in HIV testing among AGYW in Lesotho, 2004 to 2014, by age group.

### 3.2. Socio-Demographic Characteristics of Participants Using 2014 LDHS

There were 2842 AGYW participants from the Lesotho Demographic Survey conducted in 2014, with most aged between 15–19 years (52.1%), residing in urban areas (66.7%), single or never married (62.2%), and having reached higher than secondary education (52.1%). In terms of wealth index, there was an almost equal distribution between the rich and poor, with about half being either in the middle or poorer or poorest (52.3%). More than a quarter were employed (27.1%). (Table 1).

**Table 1.** Socio-demographic characteristics of AGYW using Lesotho 2014 DHS.

| Socio-Demographic Characteristics | Classification | Number of Subjects (*n*) | Proportion in Percentage (%) | 95% Confidence Interval |
|---|---|---|---|---|
| Age in Years | 15–19 | 1542 | 52.1 | 49.8–54.3 |
| | 20–24 | 1300 | 47.9 | 45.7–50.2 |
| Employment in the last 12 months | No | 2184 | 72.9 | 70.7–74.9 |
| | Yes | 658 | 27.1 | 25.1–29.3 |
| Highest level education | No education | 4 | 0.1 | 0.0–0.3 |
| | Primary | 832 | 27.7 | 25.5–30.1 |
| | Secondary | 1890 | 67 | 64.4–69.4 |
| | Higher | 116 | 52.1 | 4.0–6.8 |
| Marital status | Never in union | 1772 | 62.2 | 59.6–64.7 |
| | Currently in union | 984 | 34.6 | 32.1–37.2 |
| | Formerly in union | 86 | 3.2 | 2.6–4.1 |
| Place of Residence | Urban | 1948 | 66.7 | 30.5–36.4 |
| | Rural | 894 | 33.3 | 63.6–69.5 |
| Wealth Index | Poorest | 517 | 15.3 | 13.3–17.6 |
| | Poorer | 514 | 16.7 | 14.7–18.9 |
| | Middle | 609 | 20.3 | 18.4–22.5 |
| | Richer | 633 | 24.5 | 21.9–29.4 |
| | Richest | 569 | 23.1 | 20.3–26.1 |

*3.3. Factors Associated with HIV Testing among AGYW in Lesotho in 2014*

3.3.1. Socio-Demographic Factors

Older age (20–24 years) and being married were significantly associated with HIV testing. The odds of HIV testing were 5.5 times higher among the 20–24-year-old AGYW compared to the 15–19-year-old AGYW (88.3% vs. 57.9%, OR 5.47, 95% CI 4.26 to 7.02, $p < 0.001$). The difference remained statistically significant after adjusting for confounders (aOR 2.15, 95% CI 1.61 to 2.87, $p < 0.001$). Regarding marital status, the odds of testing for HIV were higher among AGYW in union than those who were never in union (94.4% vs. 59.1%, OR 11.58, 95%CI 8.36 to 16.04, $p < 0.001$). This remained statistically significant in the multivariable model (aOR 3.21, 95%CI 2.25 to 4.58, $p < 0.001$). Participants who were formerly married or in union had higher odds of HIV testing than those who were never married or in union (96.6% vs. 59.1%, OR 19.57, 95% CI 5.21 to 73.54, $p < 0.001$). However, the association became non-significant after adjusting for potential confounders (aOR 2.70, 95% CI 0.66 to 11.10, $p = 0.168$). We found no significant association between level of education and HIV testing. Employment status was statistically significant on univariate analysis; however, that changed following adjustment for confounders (aOR 0.86, 95% CI 0.50 to 1.49, $p = 0.598$). The odds of HIV testing were 1.8 times higher for the employed AGYW (80.6% vs. 69.5%, OR 1.83, 95%CI 1.36 to 2.46, $p < 0.001$) compared to the unemployed AGYW. (Table 2).

**Table 2.** Factors associated with HIV testing among adolescent girls and young women participants of the Lesotho 2014 DHS.

| Variables | Classification | Number of Participants | HIV Tested (%) | OR (95% CI) | *p*-Values | aOR (95% CI) | *p*-Values |
|---|---|---|---|---|---|---|---|
| | **Model** | | | *Univariate Analysis* | | *Multivariable Analysis* | |
| ***Socio-Demographic Characteristics*** | | | | | | | |
| Age in Years | 15–19 | 1542 | 57.9 | Ref | | | |
| | 20–24 | 1300 | 88.3 | 5.47 (4.26–7.02) | 0.000 | 2.15 (1.61–2.87) | 0.000 |
| Employment in the last 12 months | No | 2184 | 69.5 | Ref | | | |
| | Yes | 658 | 80.6 | 1.83 (1.36–2.46) | 0.000 | 0.86 (0.50–1.49) | 0.598 |
| Highest level of education | No education | 4 | 54.5 | Ref | | | |
| | Primary | 832 | 70.5 | 1.99 (0.27–14.96) | 0.501 | | |
| | Secondary | 1890 | 73.1 | 2.27 (0.30–17.06) | 0.425 | | |
| | Higher | 116 | 75.4 | 2.55 (0.32–20.24) | 0.374 | | |
| Marital status | Never in union | 1772 | 59.1 | Ref | | | |
| | Currently in union | 984 | 94.4 | 11.58 (8.36–16.04) | 0.000 | 3.21 (2.25–4.58) | 0.000 |
| | Formerly in union | 86 | 96.6 | 19.57 (5.21–73.54) | 0.000 | 2.70 (0.66–11.10) | 0.168 |
| Place of Residence | Urban | 894 | 70.2 | Ref | | | |
| | Rural | 1948 | 73.6 | 1.18 (0.92–1.51) | 0.182 | | |
| Wealth Index | Poorest | 517 | 71.4 | Ref | | | |
| | Poorer | 514 | 76.8 | 1.33 (0.95–1.86) | 0.097 | | |
| | Middle | 609 | 76.3 | 1.30 (0.95–1.77) | 0.103 | | |
| | Richer | 633 | 71.3 | 1.00 (0.71–1.40) | 0.995 | | |
| | Richest | 569 | 68 | 0.85 (0.62–1.17) | 0.327 | | |
| ***Risky sexual behaviour*** | | | | | | | |
| Lifetime number of sexual partners | 1 | 908 | 81.6 | Ref | | | |
| | 2 | 524 | 86.1 | 1.40 (0.98–2.00) | 0.064 | 1.03 (0.64–1.68) | 0.891 |
| | 3 or more | 493 | 90.5 | 2.14 (1.32–3.47) | 0.002 | 1.15 (0.67–1.97) | 0.613 |
| Condom use at last sexual intercourse | No | 745 | 89.7 | Ref | | | |
| | Yes | 894 | 90.5 | 0.57 (0.39–0.84) | 0.005 | 1.20 (0.70–2.08) | 0.508 |
| Had an STI | No | 1884 | 85.3 | Ref | | | |
| | Yes | 34 | 89 | 1.40 (0.35–5.71) | 0.634 | | |
| ***Knowledge about HIV*** | | | | | | | |
| Comprehensive HIV knowledge | No | 1246 | 69.6 | Ref | | | |
| | Yes | 799 | 77.6 | 1.51 (1.25–1.82) | 0.000 | 0.96 (0.62–1.49) | 0.845 |
| Knowledge of MTCT | No | 780 | 63.5 | Ref | | | |
| | Yes | 1265 | 79.6 | 2.24 (1.84–2.74) | 0.000 | 1.53 (1.21–1.94) | 0.000 |
| HIV non-discriminatory attitudes | No | 364 | 56.9 | Ref | | | |
| | Yes | 1681 | 76.8 | 2.50 (2.00–3.13) | 0.000 | 2.50 (1.87–3.34) | 0.000 |
| ***Pregnancy*** | | | | | | | |
| Ever been pregnant | No | 1012 | 57.6 | Ref | | | |
| | Yes | 1033 | 97.6 | 29.4 (19.04–45.46) | 0.000 | 11.53 (7.46–17.84) | 0.000 |
| ***Gender-Based Violence (GBV)*** | | | | | | | |
| Ever experienced GBV | No | 1201 | 74.2 | Ref | | | |
| | Yes | 844 | 70 | 0.81 (0.67–0.98) | 0.033 | 1.11 (0.73–1.68) | 0.640 |

STI, Sexually Transmitted Disease; MTCT, Mother-To-Child-Transmission of HIV; GBV, Gender-Based Violence; OR Odds Ratio; aOR, adjusted Odds Ratio.

### 3.3.2. Risky Sexual Behaviour

Study participants who had had two lifetime sexual partners had higher odds of HIV testing than those who had had one lifetime sexual partner (86.1% vs. 81.6%, OR 1.89, 95% CI: 1.20 to 2.98, *p* = 0.003). The association was no longer significant following adjustment

for potential confounders (aOR 1.03, 95%CI 0.64 to 1.68, $p = 0.891$). AGYW who had had three or more lifetime sexual partners had greater odds to test for HIV than those with one lifetime sexual partner (90.5% vs. 81.6%, OR 2.14, 95%CI 1.32 to 3.47, $p = 0.002$); however, the association was no longer significant in the multivariable model (aOR 1.15, 95% CI 0.67 to 1.97, $p = 0.613$). The odds of testing for HIV by subjects that used condoms during last sexual intercourse reduced by 43% compared to those who did not use condom at their last sexual intercourse (90.5% vs. 89.7%, OR 0.57, 95%CI 0.39 to 0.84, $p = 0.005$); this association was also not significant after adjusting for confounding (aOR 1.20, 95% 0.70 to 2.08, $p = 0.508$). Having an STI or STI symptoms was not significantly associated with HIV testing. (Table 2).

### 3.3.3. Knowledge about HIV

AGYW with comprehensive HIV knowledge had 1.5 times higher odds of testing for HIV than those without (77.6% vs. 69.6%, OR 1.51, 95%CI 1.25 to 1.82, $p < 0.001$); however, the association was no longer significant after adjusting for potential confounders (aOR 0.96, 95% CI 0.62 to 1.49, $p = 0.845$). AGYW who were knowledgeable about mother-to-child transmission of HIV (MTCT) had greater odds of HIV testing than those who did not (79.6% vs. 63.5%, OR 2.24, 95%CI 1.84–2.74, $p < 0.001$); the results remained significant after adjusting for potential confounders (aOR: 1.53, 95%CI: 1.21 to 1.94, $p < 0.001$). Among the participants, those who had non-discriminatory attitudes towards people living with HIV had higher odds of HIV testing than those who had discriminatory attitudes (76.8% vs. 56.9%, OR 2.50, 95%CI 2.00 to 3.13, $p < 0.001$). Following adjustment for possible confounders, the results remained significant (aOR 2.50, 95%CI 1.87 to 3.34, $p < 0.001$) (Table 2).

### 3.3.4. Pregnancy and Gender-Based Violence

The odds of testing for HIV by those with history of pregnancy were 29 times higher than those without history of pregnancy (97.6% vs. 57.6%, OR 29.4, 95%CI 19.04 to 45.46, $p < 0.001$). The findings remained significant after adjusted analysis (aOR 11.53, 95%CI 7.46 to 17.84, $p < 0.001$). Participants who justified gender-based violence had reduced odds for testing for HIV than those who did not (70% vs. 74.2%, OR 0.81, 95%CI 0.67 to 0.98, $p = 0.033$). After adjusting for possible confounders, the association was no longer significant (aOR 1.11, 95% CI 0.73 to 1.68, $p = 0.640$). (Table 2).

## 4. Discussion

From 2004 to 2014, the prevalence of HIV testing among AGYW in Lesotho increased by two-thirds. The rate of HIV testing increased by 50.2% over a 5-year period from 2004 and 2009, and by a mere 10.3% for the same period from 2009 to 2014 in our trend analysis. From 2009 to 2014, the rate of participants who had ever received an HIV test was very low compared to 2004 to 2009. It is likely that the overall increase in HIV testing was due to improved public awareness of HIV/AIDS and accessible HTC services. According to Peralta et al. [15], additional features that may improve HIV testing among youth include free HIV tests, rapid HIV tests, and convenient or accessible HTC services. Age was partitioned in five years age groups (15–19 and 20–24 years), and we observed that over a period of 10 years from 2004 to 2014, the uptake HIV testing increased by 51.7% amongst the 15–19 years old and 69.6% for the 20–24 years old. Progress was perceived across all age groups, but the proportion of participants who had ever received an HIV test was greater among AGYW aged 20–24 years. Generally, older youths are more educated than younger youths, and education has a remarkable impact on people's health because it empowers individuals to make healthier choices. Individuals with more years of education are likely to have better health and well-being, and tend to practice healthier behaviours [16]. The rationale behind lower rates of HIV testing among younger AGYW might thus be a lower level of education.

In 2014, the proportion of those who had ever had HIV testing for adolescent girls (15–19 years) and young women (20–24 years) was 57.9% and 88.3%, respectively. This

observation indicates inadequate uptake of HIV testing among AGYW, especially of the age group 15–19 years. UNAIDS indicated that in sub-Saharan Africa, six in seven new HIV infections among adolescents aged between 15–19 years are girls [2]; therefore, this inadequate testing is problematic given the HIV infection rate among adolescent girls. Several studies on HIV testing uptake by AGYW were carried out, and many revealed that older youth (20–24 years) have higher odds of HIV testing compared to younger ones (15–19 years). A Zimbabwean study showed that 47.9% of the AGYW aged 15–19 years tested for HIV, compared to 84.8% aged between 20–24 years [17]. Asaolu et al. [18] report that in a study performed among Tanzanian students, HIV testing for students 18 years and older was higher (48.5%) compared to those younger than 18 years (24.4%). These findings were also similar to a study in South Africa, which explored correlates of HIV testing among youth aged 15–24 years, which found that the proportion of those ever tested for HIV was 23.2% and 37.9% for age groups 15–19 years and 20–24 years, respectively. Previous studies implied lack of youth-friendly sexual and reproductive health services, societal barriers and restrictive laws and policies contributed to less uptake of HIV testing, in particular the younger youth [7,18,19]. However, there has been some development regarding restrictive laws and policies. Certain countries, including Lesotho and South Africa, have enacted less restrictive laws, under which an adolescent from 12 years is permitted to obtain HIV test without parental consent [7,18].

In the multivariable model predicting ever having an HIV test, various covariates were significant, suggesting large discrepancies in testing behaviour. These included older age, marital status (currently in a union), knowledge of MTCT, HIV non-discriminatory attitudes, and pregnancy. AGYW who were involved or with a sexual partner at the time of survey were more likely to test for HIV than respondents who were never in union. Prior research has similarly examined factors associated with HIV testing among AGYW, and has described living with a man as influencing testing behaviour [20]. One of the reputable findings is that cohabiting or married partners seldom use condoms, and that women in union, especially in patriarchal society, are at higher risk of STIs, including HIV, because of their inability to negotiate for safe sex [21,22]. Therefore, the high HIV testing rate among those in union might be attributable to frequent visitation to healthcare facilities for STI treatment, during which HIV testing may be suggested by healthcare providers. MTCT knowledge was associated with increased HIV testing. A lower HIV testing rate was identified among AGYW who lacked knowledge on MTCT. MTCT is typically linked to pregnancy, and since this study focused on young women, it is highly probable that they had been previously pregnant and enrolled on a PMTCT programme. The outstanding correlate of having been tested for HIV among AGYW was history of pregnancy, with the odds of testing for HIV among those who had ever been pregnant 11.5 times higher than those with no history of pregnancy. PMTCT was initiated in Lesotho in 2007, and HIV testing is mandatory for pregnant women to reduce the risk of vertical transmission [23]. The heightened prevalence of HIV testing among those with history of pregnancy we found could be explained by compulsory HIV testing among pregnant women. The study found that 97.6% of AGYW with history of pregnancy had tested for HIV. Other comparable studies revealed that history of pregnancy is highly associated with HIV testing [17,20,24]. HIV non-discriminatory attitude was also positively linked to HIV testing. Respondents without discriminatory behaviour towards PLHIV are more likely to contract HIV test than the discriminatory ones. The justification could possibly be that they possess comprehensive knowledge on HIV/AIDS and have no fear of stigma. Famoroti et al. [25] identified that lack of knowledge on HIV/AIDS is associated with stigmatization of PLHIV. Therefore, those with discriminatory attitudes are afraid of being stigmatized, and therefore are less likely to test for HIV.

Furthermore, the study acknowledges several other factors analyzed which appeared non-significant after adjusting for potential confounders. These factors include employment, education, marital status (formerly in a union), 3 or more lifetime sexual partners, condom use at last sexual intercourse, comprehensive HIV knowledge, and history of

gender-based violence. In contrast, Pachena et al. [17] found comprehensive HIV knowledge and having 3 or more lifetime number of sexual partners to be significantly associated with HIV testing.

This study is a secondary data analysis of DHS data, and therefore prone to reporting and recall biases which may result in underestimation or overestimation of HIV testing outcome. This is because most of the questions required retrospective data, and due to the sensitivity of the topic, reporting biases are highly likely. Moreover, there were missing data in numerous variables. For instance, variables such as gender-based violence and comprehensive knowledge were omitted, with scattered responses that required cautious interpretation and summarization. Although the DHS data have limitations, they also have strengths. The study used three LDHS data sets, of which the samples were representative of the population. Samples were large with high response rates. Another added strength is that this secondary analysis was adjusted using survey weights.

## 5. Conclusions

Overall, the uptake of HIV testing increased among AGYW in Lesotho from 2004 to 2014. However, the HIV testing rate among younger women (15–19 years) was very low. To achieve the UNAIDS goal of 95–95–95 testing and treatment targets by 2030, the utilization of HTC services needs to be scaled up. Public health strategies to surpass the current testing rate are urgently needed in Lesotho. Younger adolescents showed a trend of poor HIV testing uptake; therefore, we recommend more focus on this age group. Education and women empowerment at a tender age may play a crucial role in increasing the utilization of HTC services by AGYW. Education and campaign strategies should be aimed at females and their partners to enable sustainable support. Additionally, there is a necessity for qualitative research that assesses barriers to HIV testing uptake among adolescent girls aged 15 to 19 years old.

**Author Contributions:** Conceptualization, O.N.S. and A.M.; methodology, O.N.S. and A.M.; software, A.M.; validation, O.N.S. and A.M.; formal analysis, O.N.S.; writing—original draft preparation, O.N.S.; review and editing, O.N.S.; resources, O.N.S. and A.M.; supervision, A.M.; All authors have read and agreed to the published version of the manuscript.

**Funding:** This research received no external funding.

**Institutional Review Board Statement:** The Ethics Committee of the University of Pretoria, Faculty of Health Sciences, approved the study: 18 March 2022/Protocol code 182/2022.

**Informed Consent Statement:** The study is a secondary data analysis; therefore, informed consent for participants was not applicable.

**Data Availability Statement:** Publicly available datasets were analysed in this study. This data can be found here: https://dhsprogram.com/data/available-datasets.cfm (accessed on 27 July 2022).

**Acknowledgments:** We acknowledge the DHS Program for granting access to the Lesotho Demographic Survey data sets.

**Conflicts of Interest:** The authors declare no conflict of interest.

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
