# Peer review of "Trends and Factors Associated with HIV Testing among Adolescent Girls and Young Women in Lesotho: Results from 2004 to 2014 Lesotho Demographic and Health Surveys"

_venereology, doi:10.3390/venereology1030019_

Round 1
Reviewer 1 Report
The authors used DHS data for secondary analysis, but they did not define some of socio-demographic variables (e.g., comprehensive HIV knowledge, wealth index). They should mention how they were measured or attached supplemental information regarding the data collection tools.
They also omitted to show adjusted ORs that were not statistically significant. They should show them all. Some of the point estimates could not be statistically significant but they could be referenced by other authors who had similar or different findings.

Author Response
- Clarification on how HIV comprehensive knowledge and wealth quantile variables were measured has now been added.
- Non-significant adjusted OR have now been added both in text and table 2.
- All corrections are highlighted in red font colour in the Changes document.
Reviewer 2 Report
Thank you for submitting this manuscript for consideration. This is a very interesting article and it is well written.
Abstract - This is clear and well written. It outlines what the study is about.
Introduction - The introduction sets the scene for the study well and provides an appropriate rationale for the study. Good use of contemporary information.
Method - Overall this is well written and explains what data collection and analysis was undertaken with justification. Recommend change of sentence on line 133: It says "some few questions..." but would be better as "some questions..." Also an example may be helpful to clarify as there was no direct question on gender-based violence. In data collection, could participants answer in more than one of the data sets as these were not too far apart?
Findings - These are presented clearly and easy to understand and follow. They are well described.
Discussion - Appropriate conclusions have been drawn from the findings, making realistic recommendations
Author Response
- Changes were made to sentence line 133 as suggested.
- Examples of gender-based violence questions were added
- All corrections are highlighted in red font colour in the Changes document.